# Preservative Effects of Curcumin on Semen of Hu Sheep

**DOI:** 10.3390/ani14060947

**Published:** 2024-03-19

**Authors:** Kaiyuan Ji, Jinbo Wei, Zhiwei Fan, Mengkang Zhu, Xin Yuan, Sihuan Zhang, Shuang Li, Han Xu, Yinghui Ling

**Affiliations:** 1College of Animal Science and Technology, Anhui Agricultural University, Hefei 230036, China; kaiyuanji@ahau.edu.cn (K.J.); wjb17355380844@163.com (J.W.); fzw18714892796@163.com (Z.F.); mengkangzhu@163.com (M.Z.); yuanxin831@foxmail.com (X.Y.); sihuanzhang@ahau.edu.cn (S.Z.); lishuangcaas@163.com (S.L.); xuhan199101@126.com (H.X.); 2Anhui Provincial Key Laboratory of Local Livestock and Poultry Genetic Resources Conservation and Germplasm Innovation, Anhui Agricultural University, Hefei 230036, China; 3Linquan Comprehensive Experimental Station, Anhui Agricultural University, Fuyang 236400, China

**Keywords:** sheep, sperm, ros, metabolome

## Abstract

**Simple Summary:**

Excessive accumulation of reactive oxygen species (ROS) during semen preservation can disrupt the balance of the antioxidant system in sperm, resulting in oxidative damage to lipids and damage to their structure and function. Curcumin is a natural plant extract that neutralizes ROS and enhances the function of endogenous antioxidant enzymes. The effect of curcumin on the preservation of semen has not been reported. This study determined the effects of curcumin on refrigerated sperm (4 °C) and analyzed the effects of curcumin on sperm metabolism from a Chinese native sheep (Hu sheep). The results showed that adding curcumin significantly improved (*p* < 0.05) the viability of refrigerated sperm at an optimal concentration of 20 µmol/L. Plasma membrane and acrosome integrity in semen were significantly improved (*p* < 0.05). Adding curcumin to refrigerated semen significantly increased (*p* < 0.05) the levels of multiple antioxidant enzymes and significantly decreased (*p* < 0.05) ROS production. We identified 50 differentially expressed metabolites (DEMs) in sperm between the negative control and curcumin-treated groups, suggesting that curcumin affects ovine sperm ROS production by regulating DEMs expression. This finding provides a new way to optimize the refrigerated preservation of sheep semen.

**Abstract:**

Reactive oxygen species (ROS) are important factors that lead to a decline in sperm quality during semen preservation. Excessive ROS accumulation disrupts the balance of the antioxidant system in sperm and causes lipid oxidative damage, destroying its structure and function. Curcumin is a natural plant extract that neutralizes ROS and enhances the function of endogenous antioxidant enzymes. The effect of curcumin on the preservation of sheep semen has not been reported. This study aims to determine the effects of curcumin on refrigerated sperm (4 °C) and analyze the effects of curcumin on sperm metabolism from a Chinese native sheep (Hu sheep). The results showed that adding curcumin significantly improved (*p* < 0.05) the viability of refrigerated sperm at an optimal concentration of 20 µmol/L, and the plasma membrane and acrosome integrity in semen were significantly improved (*p* < 0.05). Adding curcumin to refrigerated semen significantly increased (*p* < 0.05) the levels of antioxidant enzymes (T-AOC, CAT, and SOD) and significantly decreased (*p* < 0.05) ROS production. A total of 13,796 metabolites in sperm and 20,581 metabolites in negative groups and curcumin-supplemented groups were identified using liquid chromatography–mass spectrometry. The proportion of lipids and lipid-like molecules among all metabolites in the sperm was the highest, regardless of treatment. We identified 50 differentially expressed metabolites (DEMs) in sperm between the negative control and curcumin-treated groups. Kyoto Encyclopedia of Genes and Genomes (KEGG) analysis revealed that DEMs were mainly enriched in the calcium signaling pathway, phospholipase D signaling pathway, sphingolipid metabolism, steroid hormone biosynthesis, 2-oxocarboxylic acid metabolism, and other metabolic pathways. The findings indicate that the addition of an appropriate concentration (20 µm/L) of curcumin to sheep semen can effectively suppress reactive oxygen species (ROS) production and extend the duration of cryopreservation (4 °C) by modulating the expression of sphingosine-1-phosphate, dehydroepiandrosterone sulfate, phytosphingosine, and other metabolites of semen. This discovery offers a novel approach to enhancing the cryogenic preservation of sheep semen.

## 1. Introduction

Refrigerated sperm is widely used to protect genetic resources and improve the efficiency of semen utilization in husbandry and biomedicine fields [1,2]. Artificial insemination of sheep primarily uses fresh semen and refrigerated semen is used much less for sheep than for pigs and cattle. Although using fresh semen yields better results, the reproductive potential of excellent RAMS breeds cannot be fully exploited because of the influence of semen’s effective time, estrous situation of ewes, and region [3]. Therefore, it is necessary to focus on the refrigerated preservation of sheep semen to develop a method for the long-term, high-quality storage of semen or a formula for semen diluents.

For low-temperature semen storage, samples are collected, diluted, and maintained at a temperature of 0–4 °C. In a low-temperature environment, the movement and metabolic activities of sperm are reduced, the accumulation of harmful metabolites is reduced, and the reproduction of harmful microorganisms, such as bacteria, is inhibited, thus extending the survival time of sperm. In the process of semen preservation, a variety of external factors, such as changes in storage temperature, semen diluent pH value, osmotic pressure changes, and the addition of various trace elements in the semen diluent type and content, affect the quality of semen preservation [4,5]. In addition, with preservation time, the metabolic activities of sperm and apoptotic sperm produce a large amount of reactive oxygen species (ROS) components [6]. When the accumulation of reactive oxygen molecules reaches a certain level, the antioxidant system of the sperm is disrupted, resulting in lipid peroxidation damage and subsequent irreversible negative effects on sperm vitality, viability, and quality [7]. Studies have shown that exogenous antioxidant substances can be added to semen diluents to reduce oxidative stress, limit the damage caused by ROS, extend sperm preservation time, and maintain quality after preservation [8,9]. Currently, the exogenous antioxidants used in semen preservation can be mainly divided into antioxidant enzymes, vitamins, amino acids, and plant extracts [10,11]. Because the mechanisms of action of different antioxidants are not the same, and the degree of response in different species is also different, there is no perfect antioxidant for semen preservation. Therefore, research on new antioxidants and the application of new antioxidant species remains ongoing.

Curcumin is a diketone compound found in the rhizomes of ginger plants that can neutralize ROS and improve the activity of endogenous antioxidant enzymes in the body [12,13]. Curcumin can also promote Nrf2 nuclear transport by inhibiting the binding of Keap1 to Nrf2, thereby upregulating the Nrf2/ARE pathway, which increases the levels of HO-1 and NQO1 and improves antioxidant activity in the body [14,15]. Curcumin has been widely used in feed additives, which can enhance the antioxidant capacity of animals and promote the nutrient absorption rate in the gut. Moreover, the production process of curcumin is mature, with low associated costs and high potential for livestock production. However, its effect on the low-temperature storage of semen has not been reported.

This study investigated the potential effects of curcumin as an antioxidant component in a semen diluent on sperm traits and antioxidant properties during the cryopreservation of Hu sheep semen. The aim is to evaluate the application potential of curcumin for the preservation of Hu sheep semen, providing a scientific basis for future research and development of preservation technology for Hu sheep semen. Notably, there is a lack of prior reports on the impact of curcumin on sheep semen preservation, highlighting the novelty and importance of this study in the field.

## 2. Materials and Methods

### 2.1. Experimental Design

In the present study, we investigated the effects of curcumin on the refrigeration of Hu sheep semen. The semen samples were collected from six healthy 1-year-old Hu sheep. Experiments were carried out to investigate the following: (1) The effect of curcumin (dilution ratio: 0, 10, 20, 40, and 80 µmol/L) on sperm quality (sperm mitochondrial membrane potential integrity, viability, and motility) under refrigeration (4 °C). (2) The effect of curcumin (dilution ratio: 0 and 20 µmol/L) on sperm membrane integrity and acrosome under refrigeration (4 °C). (3) The effects of curcumin (dilution ratio: 0 µm/L, 20 µmol/L) on antioxidant indexes of sheep sperm (T-AOC, CAT, SOD, ROS). (4) Metabolomic sequencing of Hu sheep semen (dilution ratio: 0 and 20 µmol/L).

### 2.2. Reagent Preparation

Accurately weigh 0.1 g of curcumin, dissolve it in 13.6 mL of dimethyl sulfoxide to prepare a 20 mM curcumin stock solution, and store it at −20 °C. The diluent can be a commercially available sheep long-acting diluent (BTS-Y1, Aimudo, Shijiazhuang, China).

### 2.3. Semen Collection 

Semen was collected from 6 adult male sheep using the artificial pseudo-vaginal (YJYD, Meilidun Biotechnology, Zhengzhou, China) method. Semen collection was 2 mL per sheep per day and repeated 2 times after an interval of 3 days. The ewe is fixed to the sperm collection frame, and the sperm collector holds the false vagina. The penis is introduced into the false vagina until the ram completes ejaculation, tilting the false vagina so that the semen flows into the sperm collection cup. At the end of sperm collection, semen with normal color and odor were selected, placed in an insulated sealed container, and brought to the laboratory within 1 h. In the laboratory, a small amount of semen was aspirated on a slide and examined microscopically at 37 °C. Semen with ≥85% viability was selected for testing.

### 2.4. Determination of Sperm Motility and Viability 

The sperm motility and viability were measured using a computer-assisted sperm analysis (CASA) system (McLane, ML-608JZ-H300, Scranton, PA, USA). Semen was pipetted (10 μL) on a slide, covered with a coverslip, and placed on a 37 °C thermostatic carrier table; then, five fields of view with ≥200 spermatozoa per field of view were selected. Sperm viability was defined as the percentage of total spermatozoa with motility >12 μm/s. Sperm motility was defined as the percentage of total spermatozoa with motility of 45 μm/s and moving in a straight line more than 80% of the time as a percentage of the total number of spermatozoa.

### 2.5. Determination of Mitochondrial Membrane Potential

To determine the Mitochondrial Membrane Potential (MMP), the MMP Assay Kit (Beyotime Biotechnology, Shanghai, China) was used. JC-1 is an ideal fluorescent probe for the detection of mitochondrial membrane potential. It can detect cell, tissue, or purified mitochondrial membrane potential. In this study, 100 µL of JC-1 (200×) was diluted with ultrapure water at a ratio of 1:1600. The diluted solution was added to 2 mL of staining buffer (5×) and mixed well to obtain the JC-1 staining working solution. Then, 200 µL of the test sample was resuspended in 500 µL of PBS, and 500 µL of JC-1 staining working solution was added. The samples were mixed well and incubated at 37 °C in a cell culture incubator for 20 min. During incubation, the JC-1 staining buffer (5×) with distilled water yielded JC-1 staining buffer (1×), which was kept on ice. After the 37 °C incubation period, the sample was centrifuged (600× *g*, 4 °C) for 3 min to pellet the sperm cells. The supernatant was discarded, and the pellet was resuspended in 100 µL of JC-1 staining buffer (1×). A fluorescence microplate reader (Victor Nivo, Victor, Finland) was used for detection; the JC-1 monomer has a maximum excitation wavelength of 514 nm and a maximum emission wavelength of 529 nm; JC-1 polymers (J-aggregates) have a maximum excitation wavelength of 585 nm and a maximum emission wavelength of 590 nm. Calculate data according to the manufacturer’s instructions.

### 2.6. Determination of Sperm Plasma Membrane Integrity 

Sperm low-osmotic expansion solution (1 mL) was added to a 1.5 mL centrifuge tube. The tube was preheated at 37 °C for 5 min. Then, 0.1 mL semen was added to the preheated low-osmotic expansion solution and gently shaken for mixing. The solution was incubated at 37 °C in a water bath for 30 min. Then, 10 µL of the test solution was pipetted on a glass slide and covered with a coverslip. The cells were observed under a microscope. We randomly counted 300 sperm cells, where bent-tail sperm were considered intact membrane sperm, and sperm with no changes in tail morphology were considered damaged membrane sperm. The proportion of intact-membrane sperm was calculated by dividing the number of bent-tail sperm by the total number of sperm counted. This ratio indicated the percentage of membrane integrity.

### 2.7. Determination of Sperm Acrosomal Integrity 

Under dark conditions, 10 μL of semen was placed on a slide and allowed to stand for 10 min. Then, 200 μL of 4% paraformaldehyde was used to fix the sample for 10 min. Next, 100 μL of FITC-PNA staining solution was added for acrosome staining and incubated at 37 °C in the dark for 30 min. For nuclear staining, 100 μL of DAPI staining solution was added and allowed to stand at room temperature for 5 min. The slides were washed thrice with PBS. After air-drying the slides, fluorescence microscopy (Thermo Fisher, M5000, Waltham, MA, USA) (200×) was performed. The acrosome was stained green, and the sperm nuclei were stained blue. Three random fields were selected, and two fluorescent images were captured for each field. Images with complete green cap structures represent intact acrosomes, whereas images without green fluorescence or irregular structures represent damaged acrosomes. The acrosome integrity rate was obtained by counting the proportion of intact acrosomes among the total sperm counts.

### 2.8. Determination of Sperm Total Antioxidant Capacity 

Semen samples were evaluated for antioxidant activity on days 1 and 7 of low-temperature preservation. For the control group and the group with added curcumin (20 µmol/L), 500 μL of semen was added to a 1.5 mL centrifuge tube. Centrifugation was performed (1000× *g*, 4 °C) for 5 min, the supernatant was discarded, and 500 µL of pre-chilled PBS was added at 4 °C. An ultrasonic homogenizer was used for homogenization at 4 °C before centrifugation (8000× *g*, 4 °C) for 5 min. Then, the supernatant was collected. The supernatant was diluted with pre-chilled PBS at a 4-fold dilution. A BCA assay kit (Beyotime Biotechnology, Shanghai, China, P0006) was used to determine the protein concentration. Total antioxidant capacity was measured using a total antioxidant capacity assay kit (Nanjing Jiancheng, Nanjing, China, A015-2-1). The specific steps were as follows: the ABTS working solution and peroxidase application solution were prepared according to the proportions. A standard curve was prepared using a trolox standard solution. For the tested samples, 10 µL of sample, 20 µL of peroxidase application solution, and 170 µL of ABTS working solution were added to the wells. The mixture was incubated at room temperature for 6 min, and the absorbance was measured at 405 nm using a microplate reader (Victor Nivo, Victor, Finland). Total antioxidant capacity was calculated by substituting the absorbance values of the test wells into the standard curve, multiplying by the dilution factor, and dividing by the total protein concentration.

### 2.9. Determination of Catalase (CAT) Activity 

To determine sperm catalase (CAT) activity, the CAT Assay Kit (Nanjing Jiancheng, Nanjing, China, A007-1-1) was used. Briefly, hemolysate was prepared from 50 μL of semen samples by adding double-distilled water to 5 mL. Next, 0.05 mL of the hemolysate was added to 1.1 mL of substrate (1.0 mL of reagent I and 0.1 mL of reagent II), mixed, and incubated at 37 °C for 60 s. The enzymatic reaction was terminated by adding 1.0 mL reagent III. The CAT activity was determined at 405 nm using a microplate reader (PerkinElmer Instruments Co., Ltd., Shelton, CT, USA). CAT activity is expressed in U/mL.

### 2.10. Determination of Superoxide Dismutase (SOD) Activity

The SOD Assay Kit (Nanjing Jiancheng Bioengineering Institute, Nanjing, China, A001-3-2) was used to measure sperm SOD activity. Sperm samples were centrifuged at 8000× *g* for 10 min after ultrasonication (power 20%, sonication 3 s, 10 s intervals, repeated 30 times) on ice. Then, the supernatant was collected, and an enzyme working solution and substrate solution were added before incubation. Incubation was carried out at 37 °C for 20 min. Absorbance was measured at 450 nm using a microplate reader (Victor Nivo, Victor, Finland).

### 2.11. Determination of Malondialdehyde (MDA) Content 

The MDA Assay Kit (Nanjing Jiancheng Bioengineering Institute, Nanjing, China, A003-1-2) was used to measure sperm MDA content. Sperm samples were ultrasonicated (power 20%, sonication 3 s, 10 s intervals, repeated 30 times) on ice. The sample was mixed with a pre-prepared reaction buffer reagent, boiled for 40 min, and centrifuged to collect the supernatant after cooling. Absorbance was measured at 532 nm using a microplate reader (Victor Nivo, Victor, Finland).

### 2.12. Determination of ROS

The ROS Assay Kit (Beyotime Biotechnology, Nanjing, China; E004-1-1) was used to measure the sperm ROS content. First, 200 µL of the sample was transferred to a 1.5 mL centrifuge tube and subjected to the DCFH-DA fluorescent probe 1000 times with PBS; 500 µL of diluted DCFH-DA probe was mixed with the sample to ensure thorough mixing for proper binding. The centrifuge tube was incubated at a constant temperature of 37 °C for 20 min to facilitate binding. The cells were washed thrice with serum-free cell culture medium to remove any unbound DCFH-DA. The fluorescence intensity was measured at 525 nm using a fluorescence microplate reader (Victor Nivo, Victor, Finland).

### 2.13. Purification of Spermatozoa

Sperm samples were purified using a 90–45% discontinuous Percoll gradient centrifugation method [16]. First, 0.9 mL of the 45% Percoll fraction in Sperm TALP was added to a 1.5 mL Eppendorf tube (Axygen, Corning, NY, USA), followed by the addition of 0.2 mL of the 90% Percoll fraction. The semen was carefully layered on top of the prepared Percoll gradient fraction. Next, sperm samples were pelleted by centrifugation at 950× *g* for 15 min at room temperature. The resulting sperm pellets were washed three times using phosphate-buffered saline.

### 2.14. Metabolite Extraction

The collected samples were carefully thawed on ice to ensure that the metabolites were preserved and not degraded. Metabolites were then extracted from 20 µL of each sample using 120 µL of pre-cooled 50% methanol buffer. The mixture of the metabolites and buffer was vortexed for 1 min to ensure proper mixing. After vortexing, the mixture was incubated for 10 min at room temperature to allow efficient metabolite extraction. Following incubation, the samples were stored at −20 °C overnight to extract more metabolites. The next day, the mixture was centrifuged at 4000× *g* for 20 min to separate the solid components from the supernatant containing the extracted metabolites. The supernatant was carefully transferred to 96-well plates to ensure that only the desired metabolites were transferred for further analysis. The transferred samples were stored at −80 °C prior to liquid chromatography–mass spectrometry (LC-MS) analysis to maintain metabolite stability. Furthermore, a pooled quality control (QC) sample was prepared by combining 10 µL of each extraction mixture. This QC sample served as a reference for quality assessment during LC-MS analysis.

### 2.15. LC-MS Analysis

LC-MS analysis was performed as described previously [17]. Initially, all chromatographic separations were performed using an UltiMate 3000 UPLC system (Thermo Fisher Scientific, Waltham, MA, USA). For reversed phase separation, an ACQUITY UPLC T3 column (100 × 2.1 mm, 1.8 μm, Waters, Milford, CT, USA) was utilized. The column oven was maintained at 40 °C. The mobile phase for chromatographic separation consisted of solvent A (5 mM ammonium acetate and 5 mM acetic acid) and solvent B (acetonitrile) at a flow rate of 0.3 mL/min. Gradient elution conditions were set as follows: 0 to 0.8 min, 2% B; 0.8 to 2.8 min, 2% to 70% B; 2.8 to 5.6 min, 70% to 90% B; 5.6 to 6.4 min, 90% to 100% B; 6.4 to 8.0 min, 100% B; 8.0 to 8.1 min, 100% to 2% B; and 8.1 to 10 min, 2% B.

A high-resolution tandem mass spectrometer (TripleTOF 6600; SCIEX, Framingham, MA, USA) was used to detect the metabolites eluted from the column. The Q-TOF instrument was operated in both positive and negative ion modes. The curtain gas was set at 30 PSI, Ion source gas1 at 60 PSI, Ion source gas2 at 60 PSI, and the interface heater temperature at 500 °C. For the positive and negative ion modes, the Ionspray voltage floating was set at 5000 V and −4500 V, respectively. Mass spectrometry data were acquired in the Information-Dependent Acquisition mode. The TOF mass range was set from 60 to 1200 Da. Survey scans were acquired in 150 ms, and up to 12 product–ion scans were collected if they exceeded a threshold of 100 counts per second (counts/s) with a 1+ charge state. Dynamic exclusion was performed for 4 s. During acquisition, the mass accuracy was calibrated for every 20 samples to ensure accurate mass measurements. Additionally, to evaluate the stability of the LC-MS system throughout the acquisition, a quality control sample (a pool of all samples) was acquired every 10 samples.

### 2.16. Metabolomics Data Analysis

The LC-MS data obtained were preprocessed using XCMS 3.0 software. Initially, the raw data files were converted to mzXML format and then processed using the XCMS, CAMERA, and MetaX toolboxes within the R version 4.3.2 (Eye Holes). Each ion was identified based on comprehensive information regarding retention time and m/z values. The intensity of each peak was recorded, resulting in the generation of a three-dimensional matrix that included arbitrarily assigned peak indices (retention time-m/z pairs), sample names (observations), and ion intensity information (variables). Subsequently, this information was matched with both in-house and public databases. The open-access databases Kyoto Encyclopedia of Genes and Genomes (KEGG) and HMDB were used to annotate the metabolites by matching the exact molecular mass data (m/z) with those present in the database within a threshold of 10 ppm. Additionally, the peak intensity data were preprocessed using MetaX. Features that were detected in less than 50% of quality control (QC) samples or 80% of test samples were removed. For missing peaks, values were extrapolated using the k-nearest neighbor algorithm to enhance the quality of the data. To detect outliers and batch effects, Partial Least Squares Discriminant Analysis (PLS-DA) was performed using a pre-processed dataset.

XCMS software was used to extract the signal strength information of each substance in different varieties, and metaX software (v1) was used for quality control: first, low-quality peaks were removed, and then, K-Nearest Neighbors were used to fill in missing values. Probabilistic quotient normalization and QC-robust spline batch correction were used for data normalization. This project utilized univariate analysis of fold-change and *t*-test statistical tests, followed by BH correction, to obtain q-values. Additionally, it was combined with multivariate statistical analysis using PLS-DA to determine variable importance for projection (VIP) values to identify differentially expressed metabolites (DEMs). The ratio > 2, VIP ≥ 1.0, and q value < 0.05 were set to select important features.

### 2.17. Heatmap and KEGG Analysis

We employed the heatmap package in R for the cluster analysis. Initially, we conducted log 10 data scaling and Z-score normalization of the quantitative information on the target metabolite. Subsequently, we classified the sample and metabolite expressions into two dimensions simultaneously, utilizing the Euclidean distance and average linkage connection methods. For KEGG analysis, we used Fisher’s Exact Test to compare the distribution of each KEGG pathway in the target metabolite set with that in the overall metabolite set. This allowed us to assess the significance of protein enrichment in a given KEGG pathway.

### 2.18. Statistical Analysis

Data were analyzed using GraphPad Prism software (version 8.0). Data obtained from the determination of mitochondrial membrane potential were analyzed by one-way ANOVA. Data were initially arcsine transformed prior to statistical analysis. Differences among means were assessed using the least significant difference method. Data obtained from sperm motility; viability sperm; membrane; acrosome integrity; CAT activity; SOD activity; MDA activity; T-AOC activity; and ROS activity were analyzed by two-way ANOVA. Data were initially arcsine transformed prior to statistical analysis. Differences among means were assessed using the Dunnett-t multiple comparisons test. Unless indicated, the results were expressed as the mean ± SD, and values were considered statistically significant at the *p* < 0.05 level.

## 3. Results 

### 3.1. Effects of Curcumin on Sperm Motility

The results of sperm MMP integrity detection showed a decrease with time during refrigeration. Curcumin significantly improved (*p <* 0.05) the MMP potential, and the optimal concentration was 20 µmol/L compared with the negative control group (Figure 1). The sperm viability test results showed that sperm viability decreased over time during refrigeration (Table 1). Curcumin could significantly improve (*p <* 0.05) sperm viability, and the optimal concentration was 20 µmol/L compared with the negative control group (Table 1). The results of the sperm motility test showed that sperm motility decreased over time during refrigeration. Curcumin could significantly improve (*p <* 0.05) sperm motility, and the optimal concentration was 20 µmol/L (Table 2).

### 3.2. Effect of Curcumin on Sperm Membrane and Acrosome Integrity

The sperm plasma membrane integrity was damaged as refrigeration time increased, and curcumin significantly improved (*p* < 0.05) sperm plasma membrane integrity (Figure 2A,B). The results of the MDA activity in refrigerated sperm showed a gradual decrease with prolonged storage time, and curcumin significantly increased (*p* < 0.05) MDA activity (Figure 2C). SOD activity in refrigerated sperm decreased gradually with prolonged storage time, and curcumin significantly increased (*p* < 0.05) SOD activity (Figure 2D). The sperm acrosomal integrity test results showed that sperm acrosomal integrity was damaged by prolonged storage during semen preservation. Curcumin significantly improved (*p* < 0.05) the sperm acrosomal integrity (Figure 2E,F).

### 3.3. Effects of Curcumin on Antioxidant Indexes of Sperm

The results of T-AOC activity in refrigerated Hu sheep sperm showed that T-AOC activity in refrigerated sperm gradually decreased with time of semen storage, and curcumin significantly increased (*p* < 0.05) T-AOC activity (Figure 3A). The results of CAT activity showed that the activity of CAT in refrigerated sperm decreased gradually as storage time increased, and curcumin significantly increased (*p* < 0.05) activity (Figure 3B). ROS activity in refrigerated sperm increased gradually with prolonged storage time, and curcumin significantly decreased (*p* < 0.05) the ROS activity (Figure 3C).

### 3.4. Metabolite Detection Quality Control

Partial Least Squares Discriminant Analysis (PLS-DA) of metabolites identified in sperm was performed, which revealed that metabolites in the negative control group clustered separately from those of the added curcumin groups (Figure 4). These results indicate that the addition of curcumin affected sperm metabolites.

### 3.5. Classification of Metabolites in Sperm between Negative Groups and Added Curcumin Groups

To determine the classes of all metabolites identified in sperm, the metabolites were mapped to the database. The results showed that there were 13,796 and 20,581 features in the negative and added curcumin modes annotated using the HMDB database (Figure 5). Compared with the negative group, the number of metabolites in the added-curcumin group increased. The main metabolites in the added-curcumin group were lipids and lipid-like molecules, accounting for 72.77% of all metabolites. The remaining metabolites were organoheterocyclic compounds (6.58%), organic acids and derivatives (6.3%), organic oxygen compounds (4.73%), phenylpropanoids, and polyketides (3%), followed by benzenoids and the remaining 15 metabolites (Figure 5). 

### 3.6. Effects of Curcumin on Sperm Metabolites in Negative and Added-Curcumin Groups

To directly evaluate the expression pattern of metabolites in sperm between the negative and curcumin-treated groups, a hierarchical clustering analysis of metabolites was performed. As shown in the heatmap, samples from the same groups were clustered together, and the expression patterns of metabolites exhibited differential changes between the negative and curcumin-treated groups (Figure 6A). Based on PLS-DA analyses, the variable importance in projection (VIP) value was calculated to sort the DEMs. The combined analyses of VIP > 1 and q < 0.05 revealed that the expression of 50 metabolites in sperm showed significant differences between negative and curcumin-treated groups, including 32 upregulated and 18 downregulated, were identified (Figure 6B) (*p* < 0.05). Furthermore, the top 10 metabolites with VIP > 8 included 6′,7′-dihydroxybergamottin, tetrahydrocurcumin, (-)-arctigenin, 2-(4-allyl-2,6-dimethoxyphenoxy)-1-(3,4-methylenedioxyphenyl)-1-propanol, (8R,8′R,9S)-9-hydroxy-3,4-dimethoxy-3′,4′-methylenoxy-9,9′-epoxylignan, 9-hydroxy-3′,4′-dimethoxy-3,4-methylenedioxy-9,9′-epoxylignan, 9′-epoxylignan, licoriphenone, hydroxymyricanone, and (4-{4-[2-(gamma-l-glutamylamino) ethyl]phenoxymethyl}furan-2-yl)methanamine.

### 3.7. Cluster Analyses of DEMs in Sperm between Negative and Added-Curcumin Groups

A detailed functional analysis of all DEMs in sperm between the negative control groups and curcumin-treated groups was performed using the KEGG database to screen potential biomarkers for sperm antioxidant damage. As shown in Figure 7A, primary DEMs in the sperm were mainly enriched in 15 significant metabolic pathways (*p* < 0.05), including the calcium signaling pathway, phospholipase D signaling pathway, sphingolipid metabolism, steroid hormone biosynthesis, and 2-oxocarboxylic acid metabolism, etc. (Table 3). In addition, KEGG analysis of secondary DEM (Phosphatidylglycerol) in sperm between the negative control groups and curcumin-treated groups revealed two significant metabolic pathways (*p* < 0.05), including glycerophospholipid metabolism and metabolic pathway signaling pathways (Figure 7B). Overall, the DEM-enriched metabolic pathways in the sperm are likely involved in the regulation of semen antioxidant damage.

## 4. Discussion

The incorporation of exogenous antioxidants into semen diluents has been widely discussed in the literature because of their potential to reduce the harm caused by ROS, extend storage time, and enhance sperm quality [18]. This study investigated the effect of adding different concentrations of curcumin to sheep semen samples under cold storage conditions. Statistically, the difference in sperm viability was only 3.5% difference between the highest (20 μm/L) and lowest (80 μm/L) viability on day 2 vs. 18% difference between the highest (20 μm/L) and lowest (control) on day 7. This is also the case for sperm motility. These results suggested that curcumin is aiding in the prolonged storage of sperm at 4 °C and extends the use time of fresh sperm. The addition of 20 μmol/L of curcumin was found to improve sperm vitality and viability of ram sperm, protect the integrity of sperm membrane and acrosome, suppress ROS generation, and prolong the cold storage time at 4 °C. Metabolomics showed that adding an appropriate concentration of curcumin significantly affected sphingolipid metabolism, 2-oxocarboxylic acid metabolism, and steroid hormone biosynthesis.

The plasma membrane is the outermost physiological structure of sperm. It maintains the osmotic balance inside and outside the cell, controls the transport of substances between the interior and exterior, and serves as a barrier to protect the sperm from harmful substances. Thus, plasma membrane integrity directly affects sperm viability [19]. This study confirmed that during cold storage of sheep semen, the integrity of the sperm membrane and acrosome significantly decreased with prolonged storage time. However, the addition of curcumin enhanced the integrity of the sperm plasma membrane and acrosome. The sperm plasma membrane contains a large amount of polyunsaturated fatty acids, among which the non-conjugated double bonds are highly susceptible to attack by superoxide anions, leading to lipid peroxidation (LPO) reactions, which causes the loss of fatty acids in the sperm plasma membrane, reduces membrane fluidity and integrity, and inhibits membrane proton pump function [20]. The levels of the end-products LPO and MDA reflect the degree of sperm oxidative damage [21]. Excessive MDA leads to the inhibition of the electron transport chain complex, inhibits SOD activity, and causes base-pair replacement or frameshift mutation of sperm genes, resulting in reduced genetic stability of sperm [22]. In the present study, curcumin significantly reduced MDA production and enhanced SOD activity. 

ROS are by-products of sperm metabolism or dead sperm, including superoxide anions, oxygen free radicals, and peroxides [23,24]. With prolonged cold storage of sheep sperm, the ROS content in semen continues to increase. Studies have shown that small amounts of ROS are beneficial for sperm motility and acrosomal reactions [25]. However, due to their highly reactive nature, excessive ROS accumulation can lead to sperm oxidative stress, resulting in reduced vitality and damage to the integrity of the plasma membrane, acrosome, and DNA, ultimately affecting fertilization rates [26]. This study confirmed that during cold storage of sheep semen, ROS content significantly increased with prolonged storage time. Indeed, CAT and SOD exist in sperm and seminal plasma and protect sperm from damage caused by active oxygen, and SOD catalyzes the decomposition of this ion into H_2_O_2_. Additionally, it prevents the Haber–Weiss reaction between the active oxygen, H_2_O_2_, and iron ions, resulting in the formation of highly reactive H ions [27]. Curcumin can inhibit ROS production and enhance the antioxidant capacity of animals [12,13]. However, its effect on the low-temperature storage of semen has not been reported. In this study, adding 20 μm/L of curcumin to the diluent (BTS-Y1, Aimudo, Shijiazhuang, China) significantly increased the activities of T-AOC and CAT in ram semen, which played an important role in improving the antioxidant capability of the sperm, reducing ROS production, and enhancing sperm preservation quality, which is consistent with the trend of previous studies.

The addition of curcumin improved the vitality and viability of the refrigerated sperm, thereby protecting the integrity of the sperm membrane and acrosome. It also effectively suppressed the generation of ROS, thereby prolonging the cold storage time of ram semen at 4 °C. However, the specific molecular mechanisms underlying these effects have yet to be fully elucidated. In this study, we observed that lipid molecules accounted for the greatest proportion of metabolic products in sheep sperm. Numerous studies have indicated that metabolites play critical roles in the antioxidant activity of semen [28,29]. We identified 50 DEMs, including 32 upregulated and 18 downregulated DEMs, between the negative control groups and curcumin-treated groups. Sphingosine-1-phosphate, a ceramide-induced death inhibitor, prevents sperm aging by enhancing DNA double-stranded break repair [30]. Dehydroepiandrosterone sulfate has the potential to improve mouse sperm fertilization in vitro [31]. Phytosphingosine can lead to increased ROS production in human cells [32], and the addition of curcumin to semen can effectively reduce ROS metabolism. Based on our findings, we speculate that sphingosine-1-phosphate, dehydroepiandrosterone sulfate, and phytosphingosine may serve as potential markers for reducing oxidative damage in sperm. 

## 5. Conclusions 

These results show that adding an appropriate concentration (20 µmol/L) of curcumin to sheep semen can affect sphingosine-1-phosphate, dehydroepiandrosterone sulfate, phytosphingosine, and other metabolites of semen, inhibit ROS production, and prolong the time of cryopreservation (4 °C). This finding provides a new way to optimize the refrigerated preservation of sheep semen.

## Figures and Tables

**Figure 1 animals-14-00947-f001:**
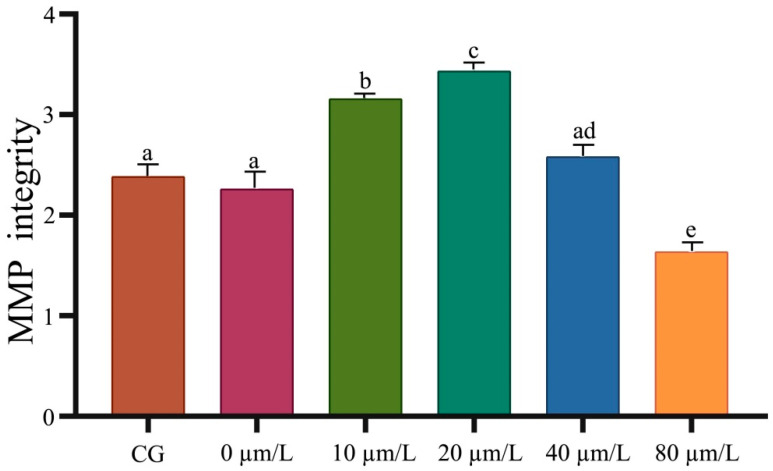
Effect of curcumin on the sperm MMP integrity in Hu sheep in refrigerated sperm. Data represent mean ± SD of the relative fold-change (n = 3 per group), and different letters on the bars indicate significant differences (*p* < 0.05).

**Figure 2 animals-14-00947-f002:**
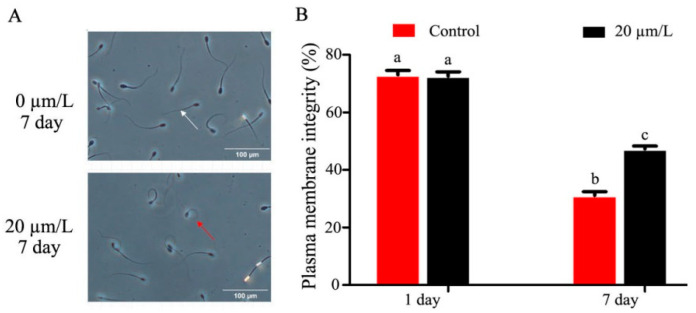
Effect of curcumin on sperm acrosome in refrigerated sperm. (**A**,**B**) Effect of curcumin on sperm plasma membrane integrity in refrigerated sperm. (**C**) Effect of curcumin on MDA in refrigerated sperm. (**D**) Effect of curcumin on SOD in refrigerated sperm. (**E**,**F**) Effect of curcumin on sperm acrosome integrity in refrigerated sperm. Data represent mean ± SD of the relative fold-change (n = 3 per group), and different letters on the bars indicate significant differences (*p* < 0.05).

**Figure 3 animals-14-00947-f003:**
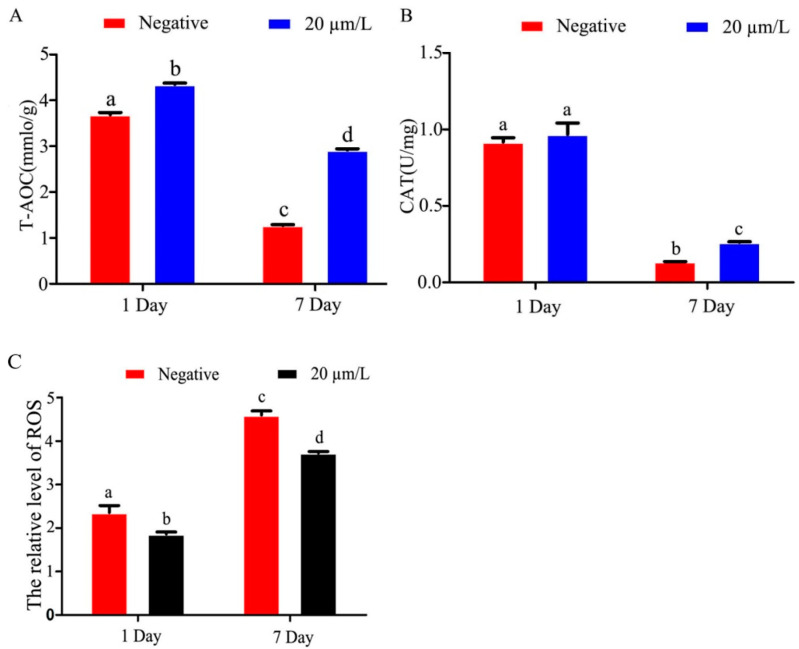
Effects of curcumin on antioxidant indexes of Hu sheep refrigerated sperm (**A**) Effect of curcumin on T-AOC in refrigerated sperm. (**B**) Effect of curcumin on CAT in refrigerated sperm. (**C**) Effect of curcumin on ROS in refrigerated sperm. Data represent mean ± SD of the relative fold-change (n = 3 per group), and different letters on the bars indicate significant differences (*p* < 0.05).

**Figure 4 animals-14-00947-f004:**
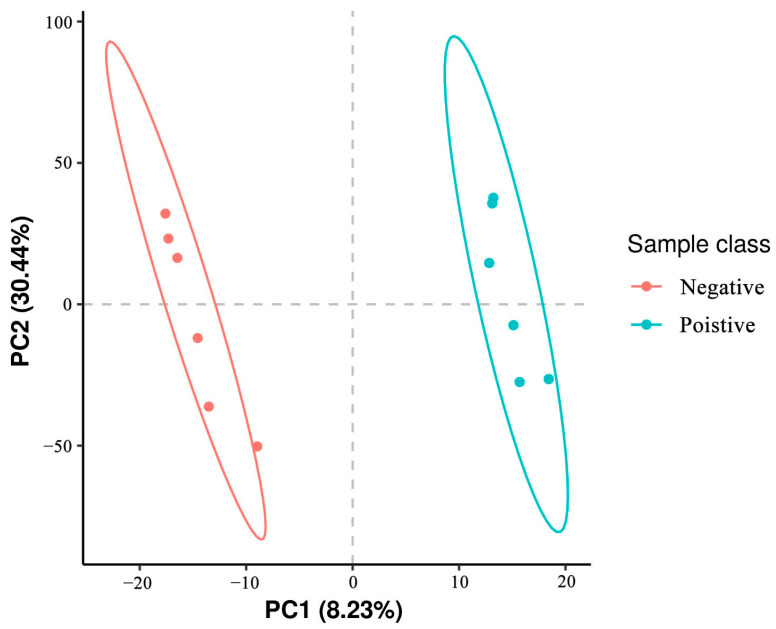
Representative images of the first (PC1) and second (PC2) principal components (PC) affecting the metabolome in sheep sperm.

**Figure 5 animals-14-00947-f005:**
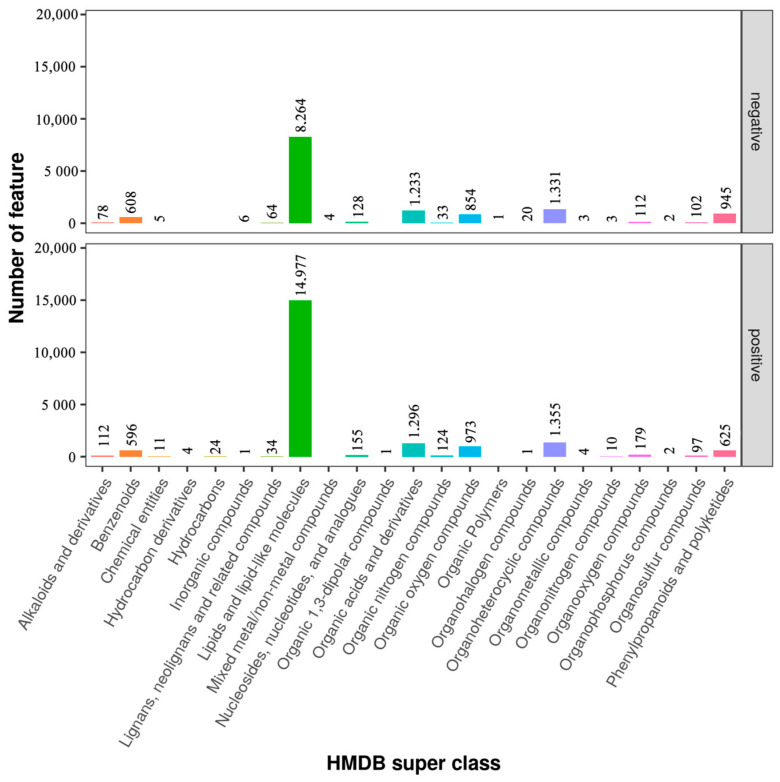
Categorization of metabolites in negative and added-curcumin groups.

**Figure 6 animals-14-00947-f006:**
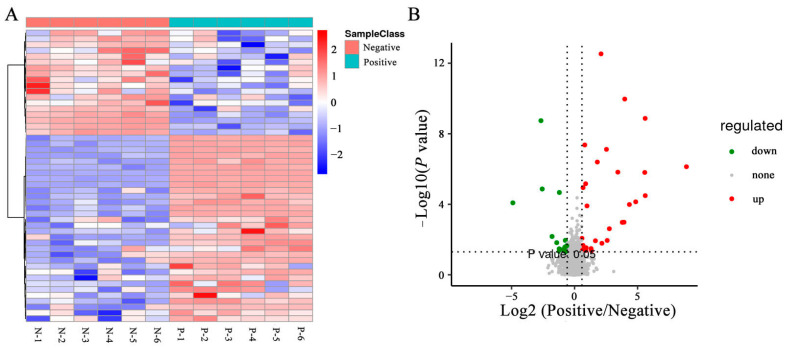
DEMs in sperm between negative and added-curcumin groups. (**A**) Heat map of DEMs between groups. (**B**) Volcano map of DEMs between groups.

**Figure 7 animals-14-00947-f007:**
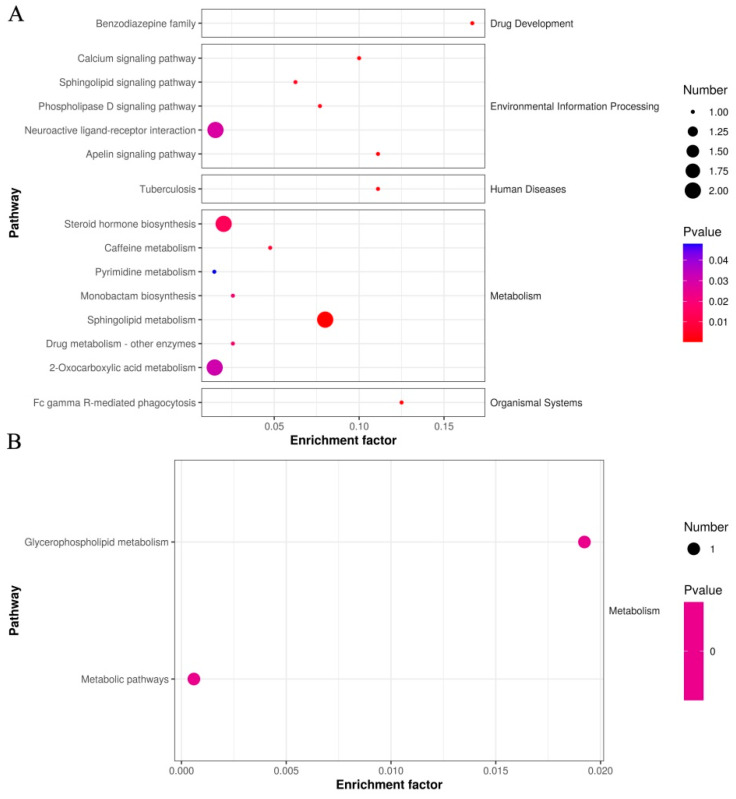
KEGG analysis of DEMs in sperm between negative and added-curcumin groups. (**A**) Differentially expressed primary metabolites annotated by the KEGG database. (**B**) Differentially expressed secondary metabolites annotated by the KEGG database.

**Table 1 animals-14-00947-t001:** Effect of curcumin on the viability of sperm of Hu sheep in refrigeration.

Time/(Day) (Day)	Control	0/(µmol/L)	10/(µmol/L)	20/(µmol/L)	40/(µmol/L)	80/(µmol/L)
1	97.86 ± 0.71	95.59 ± 0.38	97.92 ± 0.77	96.75 ± 2.31	97.87 ± 1.72	96.03 ± 1.00
2	92.68 ± 2.18 ^ab^	94.27 ± 0.71 ^a^	94.03 ± 2.58 ^a^	94.71 ± 2.72 ^a^	93.76 ± 2.61 ^ab^	90.21 ± 1.69 ^b^
3	92.11 ± 1.52 ^a^	93.93 ± 0.81 ^a^	94.19 ± 1.18 ^a^	93.38 ± 0.87 ^a^	92.95 ± 1.61 ^a^	87.88 ± 1.59 ^b^
4	80.09 ± 3.97 ^cd^	82.86 ± 0.70 ^bc^	85.99 ± 1.73 ^ab^	87.81 ± 1.28 ^a^	77.51 ± 3.05 ^de^	75.49 ± 2.73 ^e^
5	76.52 ± 1.72 ^b^	78.95 ± 0.92 ^b^	82.31 ± 1.44 ^a^	83.29 ± 1.34 ^a^	76.58 ± 1.47 ^b^	69.34 ± 1.48 ^c^
6	52.23 ± 1.58 ^d^	51.21 ± 2.16 ^d^	62.03 ± 1.84 ^b^	66.48 ± 2.07 ^a^	56.42 ± 1.35 ^c^	52.24 ± 1.59 ^d^
7	42.80 ± 3.26 ^d^	43.82 ± 3.70 ^d^	52.02 ± 3.21 ^b^	60.94 ± 2.03 ^a^	48.17 ± 2.61 ^bc^	45.15 ± 2.85 ^c^

Note: Different letters represent significant differences.

**Table 2 animals-14-00947-t002:** The effect of curcumin on sperm motility in refrigerated sperm of Hu sheep.

Time/(Day) (Day)	Control	0/(µmol/L)	10/(µmol/L)	20/(µmol/L)	40/(µmol/L)	80/(µmol/L)
1	94.00 ± 0.97	96.17 ± 1.06	94.62 ± 1.13	93.66 ± 2.91	93.69 ± 3.95	93.40 ± 1.17
2	84.54 ± 1.27 ^b^	88.74 ± 1.90 ^a^	89.31 ± 1.92 ^a^	89.37 ± 1.84 ^a^	85.75 ± 0.66 ^b^	83.24 ± 2.58 ^b^
3	80.54 ± 2.29 ^c^	84.83 ± 0.93 ^b^	84.36 ± 1.10 ^b^	86.71 ± 0.81 ^a^	81.27 ± 2.53 ^c^	81.48 ± 2.11 ^c^
4	67.71 ± 1.70 ^c^	72.54 ± 1.20 ^b^	74.87 ± 1.86 ^b^	79.27 ± 1.87 ^a^	67.05 ± 3.36 ^cd^	64.15 ± 1.85 ^d^
5	54.81 ± 2.60 ^c^	56.61 ± 2.66 ^c^	60.21 ± 1.26 ^b^	73.70 ± 2.22 ^a^	63.23 ± 2.17 ^b^	56.59 ± 2.06 ^c^
6	35.55 ± 2.09 ^c^	43.53 ± 1.02 ^c^	54.31 ± 1.46 ^b^	60.84 ± 2.30 ^a^	45.94 ± 3.95 ^c^	45.29 ± 2.82 ^c^
7	34.27 ± 2.93 ^c^	40.26 ± 2.79 ^b^	42.21 ± 2.00 ^b^	53.27 ± 1.71 ^a^	35.68 ± 3.69 ^c^	35.26 ± 2.64 ^c^

Note: Different letters represent significant differences.

**Table 3 animals-14-00947-t003:** Effects of curcumin on primary metabolites of sperm.

Metabolic Pathway	*p* Value	FDR	Compound	Compound ID
Benzodiazepine family	0.0003	0.0032	L-365260	C15026
Apelin signaling pathway	0.0009	0.0033	Sphingosine 1-phosphate	C06124
Calcium signaling pathway	0.0011	0.0034	Sphingosine 1-phosphate	C06124
Phospholipase D signaling pathway	0.0019	0.005	Sphingosine 1-phosphate	C06124
Sphingolipid signaling pathway	0.0029	0.0067	Sphingosine 1-phosphate	C06124
Neuroactive ligand-receptor interaction	0.0284	0.0394	UTP; Sphingosine 1-phosphate	C00075; C06124
Tuberculosis	0.0009	0.0033	Sphingosine 1-phosphate	C06124
Caffeine metabolism	0.0051	0.0102	Xanthosine	C01762
Monobactam biosynthesis	0.0170	0.0256	MM 42842	C20928
Metabolic pathways	0.8469	0.847	UTP; Xanthosine; Gibberellin A20; Sphingosine 1-phosphate; Cephaeline; Phytosphingosine	C00075; C01762; C02035; C06124; C09390; C12144
2-Oxocarboxylic acid metabolism	0.0307	0.0395	(R)-(Homo)2-citrate; (-)-threo-Iso(homo)2-citrate	C16583; C16597
Sphingolipid metabolism	0.0002	0.0033	Sphingosine 1-phosphate; Phytosphingosine	C06124; C12144
Steroid hormone biosynthesis	0.0138	0.0249	Dehydroepiandrosterone sulfate; 2-Methoxyestrone 3-glucuronide	C04555; C11132
Purine metabolism	0.0817	0.092	Xanthosine	C01762
Pyrimidine metabolism	0.0479	0.0575	UTP	C00075
Drug metabolism–other enzymes	0.0170	0.0256	5-Fluorouridine	C16633
Bile secretion	0.0911	0.0965	Dehydroepiandrosterone sulfate	C04555
Fc gamma R-mediated phagocytosis	0.0007	0.0033	Sphingosine 1-phosphate	C06124

## Data Availability

Data Availability Statements are available in the section www.ebi.ac.uk/metabolights/MTBLS9512 (accessed on 4 February 2024). In the sequencing data, the negative control groups were abbreviated as A-0 and the curcumin-treated groups as A-20.

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
