# Peer review of "Preservative Effects of Curcumin on Semen of Hu Sheep"

_animals, 2024, doi:10.3390/ani14060947_

Round 1
Reviewer 1 Report
Comments and Suggestions for Authors
This is an interesting study that examines the use of curcumin to improve characteristics of refrigerated (4℃) semen from Hu sheep, with the authors considering a wide range of sperm characteristics. The authors should be commended on conducting a well-designed and thorough study. Improve semen storage is important for all artificial breeding program of all species, but particular in sheep where AI success rates tend to remain lower than other production animals such as cattle. Thus, this is a highly topical line of research that is a valuable addition to ovine reproduction, particularly given that the curcumin treatment did in fact lead to improved sperm characteristics.
I do have a concern with the use of the term “cryopreserved”, as it could be slightly misleading. To me, cryopreserved refers to freezing or storing semen at a very low temperature (e.g., in liquid nitrogen or at least subzero). In this instance, I think that the term refrigerated or chilled or cold storage (a term used in the discussion) is more suitable here, as samples were stored at (4℃).
Overall, the manuscript is well written and presented. The introduction clearly outlines importance of this research and states the aims/objectives, which are appropriate given the trial design and results presented.
The methods section could be improved in places and some minor details are missing (see comments below), but otherwise the methods are described well. However, can the authors please clarify the dilution units. I assume that the unit is micromoles/L , in which case I think it should be umol/L not um/L. Please check and change throughout document as required. Also, can the authors also please clarify what the curcumin was diluted into? (i.e., what was the semen diluent for the control group?) I have some minor questions about the statistical analysis section. Was the normality of the data tested? If so, what test was used? This section could do with being a bit more descriptive. E.g., state which data were analyzed using ANOVA and which data were transformed and analyzed using Dunnett’s multiple comparison test.
The results are well presented, and the graphics/tables support the findings wells. It is stated that sperm viability decreased over time. However, in Table 1 the difference between the treatment groups also seems to have increased with time. For example, only 3.5% difference between highest (20 um/L) and lowest (80 um/l) viability on day 2, vs 18% difference between highest (20 um/L) and lowest (control) on day 7. This is also the case for table 2. I recommend commenting on this in the results section, as it supports the role of curcumin in the storage of the sperm.
The discussion covers most of the results. However, the motility and viability results could be discussed further. While it is mentioned passingly in Lines 454-455, I would recommend elaborating on this, particularly how the difference between the control and treatment groups increased over time, as this supports that curcumin is aiding in the prolonged storage of sperm at 4 degrees C. Accordingly, a statement on these motility and viability findings should also be added to the conclusion. Lastly, has there been other studies considering curcumin (or other antioxidants) to improve ovine (or other species) semen storage. If so, it would be useful to add a discuss this and why curcumin might be a preferable option. The discussion use past literature to explain the results well.
Minor comments:
Lines 55 – 57: Please add references to support this statement
Lines 101: Please check dilution units, as discussed above. Also, provide a description of the base/control semen diluent used.
Line 121-122: “Sperm viability was defined...”; this sentence could also be simplified to “Sperm viability was defined as the percentage of total spermatozoa with motility >12 um/s”.
Line 123: “Sperm motility was defined….”
Line 128: I would state “JC-1 stain….”
Lines 133 – 134: “During incubation, the JC-1 staining buffer (5X) was diluted with distilled water to produce JC-1 staining buffer (1X), which was kept on ice”
Line 137: Please provide details on the microplate reader (model and manufacturer) and wavelength used.
Line 156 – Please add microscope details and magnification used.
Line 178, 188, 196: Please ensure that all references to the microplate reader include the model + manufacturer details. (e.g., VICTOR Nivo plate reader, Perki Elmer Instruments Ltd, Shelton, CT, USA).
Line 213: Please produce details on plate reader.
Line 219: Check formatting of this unit throughout. Sometimes it is, for example, 950xg other are 950 x g
Lines 306-307: While the table is referenced in the next sentence, I still recommend add table reference to the following – “The sperm viability test results showed that sperm viability decreased over time during cryopreservation (Table 1)”. This would apply to Sperm motility too.
Line 414: Note sure about the term appropriate concentration here. I would suggest stating different concentrations or something similar. From these various concentrations you selected the most appropriate.
Comments on the Quality of English LanguageOverall, the manuscript is well written and presened.
Author Response
Reviewer: 1
This is an interesting study that examines the use of curcumin to improve characteristics of refrigerated (4 ℃) semen from Hu sheep, with the authors considering a wide range of sperm characteristics. The authors should be commended on conducting a well-designed and thorough study. Improve semen storage is important for all artificial breeding program of all species, but particular in sheep where AI success rates tend to remain lower than other production animals such as cattle. Thus, this is a highly topical line of research that is a valuable addition to ovine reproduction, particularly given that the curcumin treatment did in fact lead to improved sperm characteristics.
- I do have a concern with the use of the term “cryopreserved”, as it could be slightly misleading. To me, cryopreserved refers to freezing or storing semen at a very low temperature (e.g., in liquid nitrogen or at least subzero). In this instance, I think that the term refrigerated or chilled or cold storage (a term used in the discussion) is more suitable here, as samples were stored at (4 ℃). Overall, the manuscript is well written and presented. The introduction clearly outlines importance of this research and states the aims/objectives, which are appropriate given the trial design and results presented.
Response:Thank you very much for the experts' review of this manuscript. In the manuscript, "refrigerated" has been used to replace "cryopreserved", and the modified parts have been marked, please refer to the revised manuscript for details.
- The methods section could be improved in places and some minor details are missing (see comments below), but otherwise the methods are described well. However, can the authors please clarify the dilution units. I assume that the unit is micromoles/L , in which case I think it should be umol/L not um/L. Please check and change throughout document as required. Also, can the authors also please clarify what the curcumin was diluted into? (i.e., what was the semen diluent for the control group?) I have some minor questions about the statistical analysis section. Was the normality of the data tested? If so, what test was used? This section could do with being a bit more descriptive. E.g., state which data were analyzed using ANOVA and which data were transformed and analyzed using Dunnett’s multiple comparison test.
Response:Thank you very much for the expert advice, The author has revised the experimental methods and data statistics, and marked them in the manuscript.
2.2.Reagent Preparation
Accurately weigh 0.1 grams of curcumin, dissolve it in 13.6 mL of dimethyl sulfoxide to prepare a 20 mM curcumin stock solution, and store it at -20 °C. The diluent can be a commercially available sheep long-acting diluent (BTS-Y1, Aimudo, Hebei, china).
2.3. Semen collection
Semen was collected from 6 adult male sheep using the artificial pseudo-vaginal () method. Semen collection was 2ml per sheep per day and repeated 2 times after an interval of 3 days.
2.18. Statistical analysis
Data were analyzed using GraphPad Prism software (version 8.0). Data obtained from determination of mitochondrial membrane potential were analyzed by one-way ANOVA. Data were initially arcsine transformed prior to statistical analysis. Differences among means were assessed using the least significant difference method. Data obtained from sperm motility; viability sperm; membrane; acrosome integrity; CAT activity; SOD activity; MDA activity; T-AOC activity; and ROS activity were analyzed by two-way ANOVA. Data were initially arcsine transformed prior to statistical analysis. Differences among means were assessed using Dunnett-t multiple comparisons test. Unless indicated, the results were expressed as the mean ±SD, and values were considered statistically significant at the P<0.05 level.
- The results are well presented, and the graphics/tables support the findings wells. It is stated that sperm viability decreased over time. However, in Table 1 the difference between the treatment groups also seems to have increased with time. For example, only 3.5% difference between highest (20 um/L) and lowest (80 um/l) viability on day 2, vs 18% difference between highest (20 um/L) and lowest (control) on day 7. This is also the case for table 2. I recommend commenting on this in the results section, as it supports the role of curcumin in the storage of the sperm.
Response:Thank you very much for the expert advice, the author have revised it as suggested.
The incorporation of exogenous antioxidants into semen diluents has been widely discussed in the literature because of their potential to reduce the harm caused by ROS, extend storage time, and enhance sperm quality [18]. This study investigated the effect of adding different concentrations of curcumin to sheep semen samples under cold storage conditions. Statistically, the difference of sperm viability was only 3.5 % difference between highest (20 μm/L) and lowest (80 μm/l) viability on day 2, vs 18 % difference between highest (20 μm/L) and lowest (control) on day 7. This is also the case for sperm motility. These results suggested that curcumin is aiding in the prolonged storage of sperm at 4 ℃, and extend the use time of fresh sperm. The addition of 20 μmol/L of curcumin was found to improve sperm vitality and viability of ram sperm, protect the integrity of sperm membrane and acrosome, suppress ROS generation, and prolong the cold storage time at 4 ℃. Metabolomics showed that adding an appropriate concentration of curcumin significantly affected sphingolipid metabolism, 2-oxocarboxylic acid metabolism, and steroid hormone biosynthesis.
- The discussion covers most of the results. However, the motility and viability results could be discussed further. While it is mentioned passingly in Lines 454-455, I would recommend elaborating on this, particularly how the difference between the control and treatment groups increased over time, as this supports that curcumin is aiding in the prolonged storage of sperm at 4 degrees C. Accordingly, a statement on these motility and viability findings should also be added to the conclusion. Lastly, has there been other studies considering curcumin (or other antioxidants) to improve ovine (or other species) semen storage. If so, it would be useful to add a discuss this and why curcumin might be a preferable option. The discussion use past literature to explain the results well.
Response:Thank you very much for the suggestions of the reviewers. The author has revised the discussion part according to the suggestions of the reviewers, and the modified parts have been marked.
Minor comments:
- Lines 55 – 57: Please add references to support this statement
Response: The authors have added appropriate references as suggested, see Reference [3] for details on lines511-512.
- Lines 101: Please check dilution units, as discussed above. Also, provide a description of the base/control semen diluent used.
Response: The author have added reagents to the experimental methods, and the detail on 2.1 reagent preparation.
- Line 121-122: “Sperm viability was defined...”; this sentence could also be simplified to “Sperm viability was defined as the percentage of total spermatozoa with motility >12 um/s”.
Response: Thank you very much for the expert advice, and the author have simplified the sentence according to the suggestion on line 126-127.
- Line 123: “Sperm motility was defined….”
Response: The author have revised it as suggested, and the detail on line 127-128.
- Line 128: I would state “JC-1 stain….”
Response: The authors have added an introduction to JC-1 in the Experimental methods section.
To determine the Mitochondrial Membrane Potential (MMP), the MMP Assay Kit ( Beyotime Biotechnology, China) was used. JC-1 is an ideal fluorescent probe for the detection of mitochondrial membrane potential. It can detect cell, tissue or purified mitochondrial membrane potential.
- Lines 133–134: “During incubation, the JC-1 staining buffer (5X) was diluted with distilled water to produce JC-1 staining buffer (1X), which was kept on ice”
Response: At the end of incubation at 37 ℃, the sample needs to be centrifuged and precipitated, and the JC-1 (1X) working liquid is re-suspended. Therefore, the author prepared the JC-1 (1X) working liquid during the incubation at 37 ℃.
- Line 137: Please provide details on the microplate reader (model and manufacturer) and wavelength used.
Response: A fluorescence microplate reader (Victor Nivo, Victor) was used to detect.
- Line 156 – Please add microscope details and magnification used.
Response: After air-drying the slides, fluorescence microscopy (ThermoFisher, M5000) (200×) was performed, and the detail on line 142.
- Line 178, 188, 196: Please ensure that all references to the microplate reader include the model + manufacturer details. (e.g., VICTOR Nivo plate reader, Perki Elmer Instruments Ltd, Shelton, CT, USA).
Response: the auhor have added the manufacturer details of microplate reader, and the modified parts have been marked.
- Line 213: Please produce details on plate reader.
Response: The author have added the details on plate reader, and the modified parts have been marked, and the detail on line 221.
- Line 219: Check formatting of this unit throughout. Sometimes it is, for example, 950xg other are 950 x g.
Response: The author have revised it, and the detail on line 227.
- Lines 306-307: While the table is referenced in the next sentence, I still recommend add table reference to the following – “The sperm viability test results showed that sperm viability decreased over time during cryopreservation (Table 1)”. This would apply to Sperm motility too.
Response: The author have revised it as suggested, and the detail on line 316.
- Line 414: Note sure about the term appropriate concentration here. I would suggest stating different concentrations or something similar. From these various concentrations you selected the most appropriate.
Response: The author have revised it as suggested, and the detail on line 414.
Reviewer 2 Report
Comments and Suggestions for Authors
The manuscript is interesting. However major revision are need before publication.
The term cryopreservation refers to preservation in liquid nitrogen. In this work, however, it is indicated as conservation at 4°C. this leads the reader into confusion. The authors must therefore change the terms "cryopreservation" and "cryopreserved" to the terms "refrigeration" and "refrigerated" or “chilled”.
Materials and Methods:
The authors must indicate the CASA setting parameters. Why the author considered only the total sperm motility? for more complete results, also report other sperm motility and speed values detected by CASA system.
It is not specified in the text in the experimental design paragraph how many ejaculates were collected for each ram. the experiments should at least be carried out in triplicate to obtain reliable results. Furthermore, it was not reported after how many hours at 4°C the assessments were carried out and whether at different times (e.g. after 12-24-48 hours). Were all the reported parameters measured before and after storage? although it seems so from the results, it is not well explained in the materials and methods.
Please report which kind of extender have been use.
In the materials and methods section was not clear why the Authors carried out Percoll gradient. Please explain.
Although there are several works in the bibliography on the use of turmeric as an antioxidant for seminal material, these are not taken into consideration by the authors. Please expand the bibliography and compare what emerged from this study with what is already present in the bibliography.
Comments on the Quality of English Language
The manuscript is interesting. However major revision are need before publication.
Author Response
Reviewer: 2
- The manuscript is interesting. However major revision are need before publication.
The term cryopreservation refers to preservation in liquid nitrogen. In this work, however, it is indicated as conservation at 4°C. this leads the reader into confusion. The authors must therefore change the terms "cryopreservation" and "cryopreserved" to the terms "refrigeration" and "refrigerated" or “chilled”.
Response:Thank you very much for the experts' review of this manuscript. In the manuscript, "refrigerated" has been used to replace "cryopreserved", and the modified parts have been marked, please refer to the revised manuscript for details.
- Materials and Methods: The authors must indicate the CASA setting parameters. Why the author considered only the total sperm motility? for more complete results, also report other sperm motility and speed values detected by CASA system.
Response Many thanks to the reviewers for their suggestions. Different motility patterns of sperm involve many influencing factors. The study was designed to focus only on sperm motility and viability, corresponding to subsequent mitochondrial function. Therefore, we are sorry and regretful that we only focused on sperm motility in the statistics, and did not save other parameters of sperm movement. The effect of curcumin on ovine sperm activity will be further investigated.
- It is not specified in the text in the experimental design paragraph how many ejaculates were collected for each ram. the experiments should at least be carried out in triplicate to obtain reliable results. Furthermore, it was not reported after how many hours at 4°C the assessments were carried out and whether at different times (e.g. after 12-24-48 hours). Were all the reported parameters measured before and after storage? although it seems so from the results, it is not well explained in the materials and methods.
Response:Thank you very much for the opinions of the reviewers. The author has improved the experimental method part of the manuscript according to the opinions of the reviewers, and the detail in “2.3. Semen collection”.
Semen was collected from 6 adult male sheep using the artificial pseudo-vaginal (YJYD, Meilidun Biotechnology, Zhengzhou, China ) method. Semen collection was 2ml per sheep per day and repeated 2 times after an interval of 3 days. The ewe is fixed to the sperm collection frame, and the sperm collector holds the false vagina. The penis is introduced into the false vagina until the ram completes ejaculation, tilting the false vagina so that the semen flows into the sperm collection cup. At the end of sperm collection, semen with normal color and odor were selected, placed in a insulated sealed container, and brought to the laboratory within 1 h. In the laboratory, a small amount of semen was aspirated on a slide and examined microscopically at 37 °C. Semen with ≥85 % viability was selected for testing.
- Please report which kind of extender have been use.
Response:Semen was collected from 6 adult male sheep using the artificial pseudo-vaginal (YJYD, Meilidun Biotechnology, Zhengzhou, China ) method.
- In the materials and methods section was not clear why the Authors carried out Percoll gradient. Please explain.
Response:Sperm samples can be purified using percoll gradient. This method is referred to in a paper just published by theriogenlogy, and the authors have added corresponding references, and the detail on references 16.
- Although there are several works in the bibliography on the use of turmeric as an antioxidant for seminal material, these are not taken into consideration by the authors. Please expand the bibliography and compare what emerged from this study with what is already present in the bibliography.
Response:Thank you very much for the suggestions of the reviewers. The author has revised the discussion part according to the suggestions of the reviewers.
ROS are by-products of sperm metabolism or dead sperm, including superoxide anions, oxygen free radicals, and peroxides [23,24]. With prolonged cold storage of sheep sperm, the ROS content in semen continues to increase. Studies have shown that small amounts of ROS are beneficial for sperm motility and acrosomal reactions [25]. However, due to their highly reactive nature, excessive ROS accumulation can lead to sperm oxidative stress, resulting in reduced vitality and damage to the integrity of the plasma membrane, acrosome, and DNA, ultimately affecting fertilization rates [26]. This study confirmed that during cold storage of sheep semen, ROS content significantly increased with prolonged storage time. Indeed, CAT and SOD exist in sperm and seminal plasma and protect sperm from damage caused by active oxygen, and SOD catalyzes the decomposition of this ion into H2O2. Additionally, it prevents the Haber–Weiss reaction between the active oxygen, H2O2, and iron ions, resulting in the formation of highly reactive H ions [27]. Curcumin can inhibit ros production, and enhance the antioxidant capacity of animals [12,13]. However, its effect on the low-temperature storage of semen has not been reported. In this study, adding 20
Round 2
Reviewer 2 Report
Comments and Suggestions for Authors
The authors have made the requested corrections and have given sufficient explanations to the observations made previously.